# Study of the Performance of a Novel Radiator with Three Inlets and One Outlet Based on Topology Optimization

**DOI:** 10.3390/mi12060594

**Published:** 2021-05-21

**Authors:** Tao Zhou, Bingchao Chen, Huanling Liu

**Affiliations:** School of Electromechanical Engineering, Xidian University, Xi’an 710071, China; taozhou@stu.xidian.edu.cn (T.Z.); cbc1716192483@163.com (B.C.)

**Keywords:** topology optimization, microchannel, temperature difference, pressure drop, thermal resistance, Reynolds number

## Abstract

In recent years, in order to obtain a radiator with strong heat exchange capacity, researchers have proposed a lot of heat exchangers to improve heat exchange capacity significantly. However, the cooling abilities of heat exchangers designed by traditional design methods is limited even if the geometric parameters are optimized at the same time. However, using topology optimization to design heat exchangers can overcome this design limitation. Furthermore, researchers have used topology optimization theory to designed one-to-one and many-to-many inlet and outlet heat exchangers because it can effectively increase the heat dissipation rate. In particular, it can further decrease the hot-spot temperature for many-to-many inlet and outlet heat exchangers. Therefore, this article proposes novel heat exchangers with three inlets and one outlet designed by topology optimization to decrease the fluid temperature at the outlet. Subsequently, the effect of the channel depth on the heat exchanger design is also studied. The results show that the type of exchanger varies with the channel depth, and there exists a critical depth value for obtaining the minimum substrate temperature difference. Then, the flow and heat transfer performance of the heat exchangers are numerically investigated. The numerical results show that the heat exchanger derived by topology optimization with the minimum temperature difference as the goal (Model-2) is the best design for flow and heat transfer performance compared to other heat sink designs, including the heat exchanger derived by topology optimization having the average temperature as the goal (Model-1) and conventional straight channels (Model-3). The temperature difference of Model-1 can be reduced by 37.5%, and that of Model-2 can be decreased by 62.5% compared to Model-3. Compared with Model-3, the thermal resistance of Model-1 can be reduced by 21.86%, while that of Model-2 can be decreased by 47.99%. At room temperature, we carried out the forced convention experimental test for Model-2 to measure its physical parameters (temperature, pressure drop) to verify the numerical results. The error of the average wall temperature between experimental results and simulation results is within 2.6 K, while that of the fluid temperature between the experimental and simulation results is within 1.4 K, and the maximum deviation of the measured *Nu* and simulated *Nu* was less than 5%. This indicated that the numerical results agreed well with the experimental results.

## 1. Introduction

With the development of science and technology, miniaturized and integrated electronic devices have emerged, which has led to higher heat flux. The resulting high heat flux may lower the performance of electronic devices and even reduce the life of electronic components. Therefore, the heat dissipation of such a high flux in electronic components has become a research focus. According to research speculation, the heat flux of electronic devices will increase sharply and may exceed 1000 W/cm^2^ [1]. Traditional air cooling can no longer meet the heat dissipation requirements for these electronic components with high flux. In recent years, miniaturized, light-weight, liquid-cooling microchannel technology having a heat dissipation capacity up to 400 W/cm^2^ [2] has been used.

In order to improve the dissipation of the heat generated by these components, the shape design of the liquid-cooling microchannels has been studied [3]. Wang et al. [4] proposed a new type of louvered microchannel radiator and studied the convective heat transfer capacity numerically and experimentally. Lorenzini [5] proposed T–Y assembly of fins based on a structural design for heat removal. Chen et al. [6] designed a fractal tree-like microchannel radiator and conducted the experiment, reporting that the thermal efficiency was greatly improved. To lower the thermal resistance in the system, Biserni et al. [7] designed an H-shaped flow channel. They reported that the performance of the new heat sink was superior to that of conventional heat exchangers, including T-type, C-type and rectangular microchannel heat exchangers. Experimental and numerical studies were conducted on multiple microchannels [8,9]. Their designs included straight channels, serpentine channels and U-shaped channels with a countercurrent. Liu et al. [10] proposed a novel annular microchannel radiator. Xie et al. [11] examined the cooling capacity of a wave-shaped microchannel radiator while reporting that the new wave-shaped microchannel radiator had better heat transfer performance. At the same time, the hybrid method overcame the limitations of the existing numerical methods and experimental methods [12,13,14], which led to research of heat exchangers reaching a certain high level.

All the microchannel structures mentioned above were conceived by the designers. Moreover, the optimal geometric parameters were also studied, and the structures improved the heat transfer performance of the microchannel to some extent. However, due to conceptual limitations, design flexibility in reducing the internal resistance of microchannels and the uniform surface temperature is lacking.

Topology optimization abandons the limitation of size optimization to give higher design freedom, wider design space and higher flexibility. Yao et al. [15] explored the influence of the heat transfer weighting coefficient on the flow channel structure in topology optimization and concluded that the larger the coefficient, the more curved the flow channel structure. The heat sink was designed by topology optimization under natural convection with variable heat transfer coefficient boundary conditions [16,17]. The results showed that design has lower thermal resistance and a smaller mass. Zhao et al. [18] proposed a plan to optimize the internal passage and the layout of the inlet and outlet of the heat sink. Zeng et al. [19] carried out topology optimization with the goals of minimum pressure drop and minimum temperature, and they analyzed the performance by numerical simulations. Gao et al. [20] used the modified bidirectional evolutionary structural optimization method to solve the topology optimization of steady-state heat conduction problems under design-independent and design-dependent thermal loads. Lv et al. [21] proposed a new material interpolation method that was used to study the influence of parameters on the optimization process. Iradukunda et al. [22] forwarded the best heat dissipation structure by topology optimization with phase change materials (PCM). Zhang et al. [23] presented 2D and 3D nanofluid-cooled heat exchangers designed by topology optimization. Hu et al. [24] studied the flow and heat transfer characteristics of topological microchannel heat sinks under an uneven heating flux. Lei et al. [25] proposed the use of a 3D stereolithography printing pattern investment casting to manufacture 3D metal heat transfer devices, thereby further proving that topology optimization design always has better performance. Deng et al. [26] studied the multi-outlet flow distribution problem by using topology optimization to minimize fluid resistance. Liu et al. [27] solved the topology optimization problem for a multi-outlet heat exchanger.

Substrate temperature uniformity has emerged as an important factor in heat sink design. This may also be achieved by interruption technologies that promote a thermally developed flow [28] and modify inlet and outlet arrangements [29]. Notably, the parameter optimization of those heat exchangers is based on the predicted structure, so the optimal effect is limited. In contrast, topology optimization can generate a new type of heat sinks that is not predicted. Moreover, the heat sinks with multi-inlet and single-outlet arrangements achieved by topology optimization have not been examined widely. Furthermore, the effect of channel depth on thermal performance is significant and that of the channel depth during topology optimization has not been studied. Hence, we carried out the optimization for the design domain with a multi-inlet and single outlet arrangement which further decreased the maximum temperature to obtain interruptions for achieving heat transfer improvement and pressure drop reduction. The flow and thermal performance of the optimized minichannel radiators (Model-1 and Model-2) was numerically studied and compared with the conventional heat sinks in Model-3. Experimental verification was carried out for Model-2 at last.

## 2. Topology Optimization Procedure

We generated a new kind of minichannel heat sink and compared its thermal performance with that of conventional heat sinks. How we used density method in the design problem of topology optimization is introduced in the following Sections.

### 2.1. Topology Optimization Model Establishment

As shown in Figure 1, the design domain is the square area (*L* × *L*) with three entrances and one exit. The entrances are arranged at the left, middle and right of the rectangular domain. The width of the left entrance is *W*_l_, the width of the right one is *W*_r_, and the width of the middle one is *W*_m_. The exit with the width of *W*_o_ is located at the left side of the lower end. Table 1 shows the design parameters.

### 2.2. Topology Optimization for Fluid Field

In this paper, the incompressible laminar Newtonian fluid at a low Reynolds number was adopted for topology optimization. Moreover, the Reynolds number can be obtained:(1)$$Re=\frac{\rho uD_{h}}{\mu }$$
(2)$$D_{h}=\frac{2W\cdot H}{W+H}$$
where *μ* and *ρ* refer to dynamic viscosity and density, respectively; *u* is the fluid velocity; and *D*_h_ refers to the characteristic length at the inlet.

The governing equations with boundary conditions in the flow field can be set up
(3)$$\rho \left(u\cdot \nabla u\right)=-\nabla P+\nabla \{\mu (\nabla u+\left(\nabla u{)}^{T}\right)\}+f$$
(4)$$u=u_{D}\;on\;\Gamma_{D}$$
(5)$$n(-pI+\mu \left(\nabla u+\left(\nabla u{)}^{T}\right)\right)=0\;on\;\Gamma_{N}$$
where ∇ and *P* are the gradient operator and the pressure, respectively; *f* is the volume force; and *n* represents the normal vector.

The continuity equation can be written as
(6)ρ∇⋅u=0

The boundary conditions are the same as the actual conditions.
(7)uin=Re⋅μρ⋅Dh
(8)$$P_{\mathrm{out}}=1atm$$

However, the phase of every discretized element in the design domain is represented by *γ*, which is employed to unify fluid and solid, the principle of topology optimization. Making the velocity approach zero at the solid material region while analyzing fluid flow in porous media, we modify Equation (3) as
(9)$$\rho \left(u\cdot \nabla u\right)=-\nabla P+\nabla \{\mu (\nabla u+\left(\nabla u{)}^{T}\right)\}-\alpha u$$
where *α* is the resistance coefficient of the porous medium.

### 2.3. Topology Optimization for Thermal Field

In the process of topological optimization, solid and liquid phases coexist. Therefore, the fluid–solid conjugate heat transfer equation and boundary conditions are shown below:(10)$$\tilde{\rho }\tilde{C}\left(u\cdot \nabla T\right)=\nabla \cdot \left(\tilde{k}\cdot \nabla T\right)+Q$$
(11)$$T=T_{D}\;on{\;\Gamma }_{D}$$
(12)$$-n\cdot \tilde{k}\nabla T=0\;on{\;\Gamma }_{N}$$
where $\tilde{\rho }$, $\tilde{C}$ and $\tilde{k}$ refer to density, specific heat capacity and thermal conductivity of porous medium, respectively; *T* is the fluid temperature; *Q* stands for the bulk heat source, and *n* is the unit normal vector outward from *T*. The inlet temperature is constant, and the outlet is the pressure outlet. The density, specific heat capacity and thermal conductivity of porous media are described in detail in the subsequent sections.

If the velocity is zero in Equation (10), the heat transfer equation in the solid domain can be obtained:(13)$$0=\nabla \cdot \left(\tilde{k}\cdot \nabla T\right)+Q$$

### 2.4. Topological Optimization Material Interpolation Function

The Brinkman penalty model is adopted to achieve zero velocity in the solid domain. This model was first introduced into fluid topology optimization by Borrvall and Petersson [30]. As mentioned above, the choke coefficient α is used to control the solid and liquid distribution in the design domain, which is embodied in the following formula:(14)f=−αu

When *α* equals zero, the frictional force is zero. This represents the fluid region. When the value of *α* is infinite, the velocity tends to zero. This refers to the solid region. Furthermore, the resistance coefficient, density, specific heat capacity, and thermal conductivity are associated with design variables to control the local permeability of the medium. In this paper, convex functions are used to interpolate the resistance coefficient, density, specific heat capacity and thermal conductivity coefficient, respectively. These coefficients can be evaluated by
(15)$$\left\{\begin{array}{l} \tilde{\alpha }=\alpha_{s}+\left(\alpha_{f}-\alpha_{s}\right)\gamma \frac{1+p_{\alpha }}{\gamma +p_{\alpha }}\\ \tilde{k}=k_{s}+\left(k_{f}-k_{s}\right)\gamma \frac{1+p_{k}}{\gamma +p_{k}}\\ \tilde{\rho }=\rho_{s}+\left(\rho_{f}-\rho_{s}\right)\gamma \frac{1+p_{\rho }}{\gamma +p_{\rho }}\\ \tilde{C}=C_{s}+\left(C_{f}-C_{s}\right)\gamma \frac{1+p_{C}}{\gamma +p_{C}} \end{array}\right\}$$
where *γ* is the design variable ranging from zero to unity. For *γ* = 1, the design domain is the fluid domain. However, for *γ* = 0, the design domain is the solid domain. The penalty factor is represented by *p*; *p*_α_ represents the resistance coefficient; *p**_k_* represents the thermal conductivity penalty factor; *p**_ρ_* is the density penalty factor; and *p_C_* stands for the specific heat capacity penalty factor.

### 2.5. Topology Optimization Objective Function

There are several criteria regarding the heat transfer performance of radiators in topology optimization, such as maximum temperature, minimum average temperature and minimum temperature difference. In this paper, the minimum average temperature and the minimum temperature difference were used as objective functions. The expressions are listed as follows:(16)$$R_{T}=\frac{1}{V}\int_{\Omega_{d}}{\left(T-\tilde{T}\right)}^{2}d\Omega_{d}\;\tilde{T}=\frac{1}{V}\int_{\Omega_{d}}Td\Omega_{d}$$
where *R_T_* stands for the temperature difference between the design domain; *V* represents the area of the topology optimization design domain; and $\tilde{T}$ is the average temperature of the heat source.

In addition, we used the dissipating power to represent the flow resistance. Therefore, the third optimal target was the dissipating power *Φ* of the fluid, which can be expressed as follows:(17)$$\Phi =\int_{\Omega }\frac{1}{2}\mu \underset{i,j}{\sum }{\left(\frac{\partial u_{i}}{\partial x_{j}}+\frac{\partial u_{j}}{\partial x_{i}}\right)}^{2}+\underset{i}{\sum }\alpha u_{i}{}^{2}d\Omega$$

Finally, we used the weighted coefficient *w* to combine the temperature field and fluid field.

Therefore, the objective function *Ψ*_1_ which is used to couple the temperature difference and dissipating power, and the objective function *Ψ*_2,_ which is used to couple the average temperature and dissipating power, can be obtained:(18)$$\Psi_{1}=wVR_{T}+\left(1-w\right)\Phi$$
(19)$$\Psi_{2}=w\tilde{T}+\left(1-w\right)\Phi$$

Each objective function needs to be expressed in a dimensionless form. Therefore, we have
(20)$$VR_{T}^{\prime }=\left(VR_{T}-VR_{T}^{\left(\min\right)}\right)/\left(VR_{T}{}^{\left(0\right)}-VR_{T}^{\left(\min\right)}\right)$$
(21)$${\tilde{T}}^{\prime }=\left(\tilde{T}-{\tilde{T}}^{\left(\min\right)}\right)/\left({\tilde{T}}^{\left(0\right)}-{\tilde{T}}^{\left(\min\right)}\right)$$
(22)$$\Phi^{\prime }=\left(\Phi -\Phi^{\left(\min\right)}\right)/\left(\Phi^{\left(0\right)}-\Phi^{\left(\min\right)}\right)$$
where the *VR_T_*^(0)^, *T*^(0)^ and *Φ*^(0)^ refer to the initial value of temperature difference (average temperature and fluid dissipation, respectively). *VR_T_*^(min)^, *T*^(min)^ and *Φ*^(min)^ refer to the minimum value of temperature difference (minimum value of average temperature and minimum value of fluid dissipation, respectively). This paper took the initial value as the maximum value of the target normalization. As a result, the final mathematical model is as follows:(23)$$\mathrm{Find}\;\gamma_{i}\left(i=1,2,\ldots ,N\right)\;\mathrm{Minimize}\;\prod_{1}=wVR_{T}^{\prime }+\left(1-w\right)\Phi^{\prime }\;\prod_{2}=w{\tilde{T}}^{\prime }+\left(1-w\right)\Phi^{\prime }\;\mathrm{Subject}\;\mathrm{to}\;\{\begin{array}{l} Eqs.\;(3)-\left(9\right)\\ \int_{\Omega }\gamma \left(x\right)d\Omega \leq f_{v}\int_{\Omega }1d\Omega \\ 0\leq \gamma_{i}\leq 1\left(i=1,2,\ldots ,N\right) \end{array}$$
where $f_{v}$ is the volume fraction in the upper-limit constraint of fluid volume.

### 2.6. Topology Optimization Solution

In Section 2.5, we established the mathematical model for topology optimization. In this part, a solution is analyzed for this model. To solve the problem of fluid—solid conjugate heat transfer topology optimization model, the following steps are generally required:Step 1: Input model data and construct model design domain.Step 2: Discretize the structure design domain.Step 3: Conduct finite element analysis of flow field and temperature field.Step 4: Estimate whether the objective function converges.Step 5: If it converges, go to step 6; if not, sensitivity analysis and optimization algorithms are used to update the design variables and return to step 3.Step 6: Output the optimal model (the specific flowchart is shown in Figure 2).

The topology optimization process is carried out by COMSOL Multiphysics 5.4(COMSOL; Sweden; 2018). Among them, the standard adjoint variable method is used to perform the sensitivity analysis [31]. The optimization method uses the Method of Moving Asymptotes (MMA) [32]. To solve the finite element problem, we solved the flow field problem first; then, we transferred the velocity obtained from the flow field to the temperature field, and finally solved the problem of the temperature field. After obtaining the new value of hydrodynamic viscosity, we again transferred the new value to the flow field and carried out a new round of flow-field solutions until it converged. At the same time, to avoid network dependence and chessboard problems, we used HTPDEF (Helmholtz type partial differential equation filter) [33]. The HTPDEF is as follows:(24)$$-R_{f}^{2}\nabla^{2}\tilde{\eta }+\tilde{\eta }=\eta$$
(25)$$\gamma =H\left(\tilde{\eta }\right)$$
(26)$$n\cdot \left(R_{f}^{2}\nabla \tilde{\eta }\right)=0\;on{\;\Gamma }_{N}$$
where *R_f_* is the radius of the Helmholtz filter; *ῆ* is a new variable generated by filtering; *η* is a function which ranges from 0 to 1; *n* is the outward normal vector of the boundary Γ*_N_*; and $S\left(\tilde{\eta }\right)$ is expressed as the regularized Heaviside function:(27)$$S\left(\tilde{\eta }\right)=\{\begin{array}{l} 0\;(\tilde{\eta }<-h_{t})\\ \frac{1}{2}+\frac{15}{16}\left(\frac{\tilde{\eta }}{h_{t}}\right)-\frac{5}{8}{\left(\frac{\tilde{\eta }}{h_{t}}\right)}^{3}+\frac{3}{16}{\left(\frac{\tilde{\eta }}{h_{t}}\right)}^{5}\;\left(-h_{t}\leq \tilde{\eta }\leq h_{t}\right)\;\\ 1\;(h_{t}<\tilde{\eta }) \end{array}\}$$
where *h_t_* is a positive parameter of bandwidth lying between the complete fluid domain (*h_t_* < $\tilde{\eta }$) and the complete solid domain ($\tilde{\eta }$ < *h_t_*); Therefore, the bandwidth equals 2 *h**_t_*. By tightening bandwidth, we can clarify boundaries that are largely *R_f_* obscured [33].

### 2.7. Topology Optimization Simulation

In the first section, we established the geometric model for topology optimization. This section performs topology optimization for the model. The initial temperature is 293 K, and the boundaries and parameter settings are shown in Table 2. For multiphysics analysis, the flow analysis is controlled by Equations (3)–(6) and Equation (8), and the heat transfer analysis is controlled by Equations (9)–(12). In addition, the material parameters are shown in Table 3. Through calculating, the topological shape of the runner can be obtained. Figure 3 shows the topology optimization contours for the lowest average temperature as the goal, and Figure 4 gives the topology optimization picture for the lowest temperature difference as the goal.

We can see from Figure 4 that better uniform flow distribution and temperature were observed by the smallest temperature difference goal. The fluid in Figure 4 passed through as many areas as possible in the design domain, so that temperature uniformity improved. On the other hand, Figure 3 shows reductions in the average temperature for the given region on the heat source surface, while ignoring the uniformity of temperature in the design domain, so the fluid-solid contact area is relatively small.

## 3. Three-Dimensional Numerical Model

To verify the superiority of the topology optimization model for heat dissipation, this paper used SOLIDWORKS 2019 software (Dassault, Waltham, MA, USA, 2018) to set up the three-dimensional models (shown in Figure 5), which are stretched from the topological two-dimensional model. Then, the heat sink derived by topology optimization is analyzed by ANSYS FLUENT15.0 (ANSYS, Canonsburg, PA, USA, 2013). Figure 6 shows the three views of Model-3.

Figure 5a is the Model-1, which used the smallest average temperature as the objective function. Figure 5b is Model-2, which used the smallest temperature difference as the goal. At the same time, Model-3 (Figure 6) was established for comparison of the heat transfer rate with the two models derived by topology optimization. The volume fraction of the fluid is 0.5, and the size and layout of inlet and outlet are the same as the Model-1 and Model-2. The three views of Model-3 are in Figure 7. The dimensions are listed in Table 4.

### 3.1. Numerical Simulation Control Equation and Setting of Boundary Conditions

The cooling capacity of the heat sink is studied by numerical simulations. The related governing equations and the corresponding boundary conditions are described to solve this problem.

#### 3.1.1. Governing Equation

When performing numerical simulations, we make the following assumptions:(1)The fluid flow and heat transfer are stable;(2)The fluid is incompressible;(3)The physical parameters of the materials are constant;(4)The effects of gravity and external radiation are ignored;(5)The volume force and surface tension are not considered.

The governing equations are as follows:(28)ρ∇⋅u=0
(29)$$\rho \left(u\cdot \nabla u\right)=-\nabla P+\mu \nabla^{2}u$$

The energy equation for the fluid is:(30)$$\rho C_{p}\left(u\cdot \nabla T\right)=k_{f}\nabla^{2}T$$

The energy equation for the solid is:(31)$$k_{s}\nabla^{2}T=0$$

#### 3.1.2. Boundary Conditions

The velocity at the inlet is constant, the pressure at the outlet is at atmospheric pressure, and the initial fluid temperature is 293 K. The constant heat flux is applied to the bottom of the heat sink, while the other surfaces of the heat sink are insulated.

The pumping power *P_p_* can be determined as
(32)$$P_{p}=\Delta P\cdot Q_{v}$$
where Δ*P* represents the pressure drop between the inlet and outlet. The volume flow rate *Q_v_* can be determined by
(33)$$Q_{v}=Q_{m}/\rho$$
(34)$$Q_{m}=u_{\mathrm{in}}\cdot W\cdot H\cdot \rho$$
where *Q_m_* is the mass flow rate; *u*_in_ is the inlet flow rate; and *W* and *H* indicate the length and the width of the inlet.

The average temperature of the fluid *T_g_* can be evaluated as
(35)$$T_{g}=\frac{T_{\mathrm{in}}+T_{\mathrm{out}}}{2}$$
where *T*_in_ represents the inlets temperature of fluid, and *T*_out_ represents the outlet temperature of the fluid.

The Nusselt number *Nu* can be defined as
(36)$$Nu=\frac{qD_{h}}{k_{f}\left(T_{bar}-T_{g}\right)}$$
where *k**_f_* is the thermal conductivity of the fluid, and *T_bar_* represents the average temperature of the heat source surface.

The thermal resistance of the heat sink can be expressed as:(37)$$R_{th}=\frac{T_{bax}-T_{in}}{q\cdot L_{W}\cdot L_{W}}$$
where *T_bax_* represents the maximum temperature of the heat source surface of the radiator.

The Reynolds number and volume flow rate corresponding to the speed are shown in Table 5. The heat flux *q* is 50 KW/m^2^.

### 3.2. Grid Independence Verification

ANSYS-CFD-ICEM 15.0 was used to produce grids for these models. Furthermore, the mesh quality needed to be greater than 0.25 to meet the requirements of numerical calculation accuracy in the laminar region. To eliminate the influence of mesh quality on the numerical simulation, we conducted grid independence tests. In this work, Model-1 used three different grid systems: 4,384,744, 9,561,795, and 13,758,797 for testing. Model-2 used three different grid systems (5,536,079, 9,561,361, 18,450,641) as did Model-3 (5,541,460, 9,569,202, 13,299,555). Table 6 lists the maximum substrate temperature *T_max_* and pressure drop Δ*P* at different grid numbers. The table shows that the relative error of *T_max_* and Δ*P* did not exceed 0.021% and 3.417%, respectively. Therefore, all three models used a medium number of grids.

The grids of Model-2 and Model-3 are shown in Figure 8. The tetrahedral unstructured grids were adopted, and the mesh at the inlet and outlet surfaces was encrypted.

### 3.3. Analysis of Simulation Results

Through simulation, we derived the temperature field contours for different radiators at *u* = 0.18 m/s as shown in Figure 9.

Figure 9a shows the temperature field distribution of Model-1. It can be seen from the figure that the inlet and outlet temperatures of the model were relatively low, and the temperature in the middle of the left side for this radiator was relatively high because no fluid passed through the middle part on the left side, thus leading to a higher temperature in this region.

Figure 9b shows the temperature field distribution of Model-2, which was more uniform. This is because the flow was evenly distributed, and the heat transfer area became larger, leading to the decrease in the spot temperature.

Figure 9c shows the temperature field distribution of Model-3, and we can see that a hot spot occurred near the right side of the radiator. Moreover, the temperature difference of Model-3 is larger than those of models obtained by topology optimization. The reason is that this part had no fluid passed though these locations, thus leading to a higher temperature.

The variation in temperature difference with the Reynolds number is shown in Figure 10. We found that the temperature difference decreased as the Reynolds number increased for all three models. The temperature uniformity of Model-2 was always the best. Compared with Model-3, when the Reynolds number was 950, the temperature difference of Model-1 was reduced by 37.5%, and that of Model-2 by 62.5%.

The variation of the average temperature of the heat source with the Reynolds number is shown in Figure 11. When the Reynolds number increased, the average temperature decreased for all three models. As the Reynolds number (i.e., the flow rate) increased, the heat transfer coefficient increased, and the average temperature of the heat source surface of the radiator decreased. Compared with Model-3 when the Reynolds number was 950, the substrate average temperature of the Model-1 was reduced by 5.80%, and the average substrate temperature of Model-2 was reduced by 3.88%.

The relationship between the Reynolds number and convective thermal resistance is shown in Figure 12. The thermal resistance for all three models decreased as the Reynolds number increased. Under the same Reynolds number, Model-2 had the smallest thermal resistance. When the Reynolds number was 1515, the thermal resistance of Model-1 was reduced by 21.86% and that of Model-2 was decreased by 47.99% compared to Model-3. This showed that Model-2 had the strongest heat transfer ability, due to the improvement of mixing between the fluid and channel wall, as well as the smallest thermal resistance.

## 4. The Influence of Channel Depth on the Performance of Heat Exchanger

Topology optimization for the channel depths (4, 6, 8, 10 and 12 mm) was performed by COMSOL. Figure 13 shows that some differences existed in the material distribution contours of heat exchangers for different depths. Three-dimensional models were then obtained by stretching these material distribution diagrams as shown in Figure 14. The distribution for the depth of 4 mm was similar to that of 8 mm. The solid part to the right side of the material distribution diagram was interrupted compared to other depths. This may have been the result of a stronger heat exchange ability.

The thermal performances of heat exchangers for the channel depths of 4, 6, 8, 10 and 12 mm with the same height of the exchanger were studied numerically. The temperature contours are shown in Figure 15. The heat exchanger with a depth of 8 mm had better temperature uniformity. This may be attributed to the growth of flow mixing. When the inlet velocity was 0.18 m/s, the temperature difference of the bottom surface of the heat exchangers with different depths is shown in Figure 16a. The results showed that the bottom surface of the heat exchanger with a channel depth of 8 mm had the minimum temperature difference. This is the critical depth.

Figure 16b shows the effect of the depth on thermal resistance. It can be seen that as the depth of the heat exchanger channel increased, the thermal resistance gradually decreased. This can be explained by the fact that both of the convective and conductive thermal resistances were reduced. Figure 16c indicates that the variation of the Nusselt number with the channel width. The growth of Nusselt number was found when the channel depth increased because the heat transfer area accelerated. Figure 16d shows the effect of depth on the pressure drop, which decreased with increasing channel depth because of the enlarged hydraulic diameter.

## 5. Experimental Tests

To verify the accuracy of the numerical results of the radiator derived by topology optimization, we carried out experimental verification for Model-2.

### 5.1. Experimental System Design

The complete test platform consisted of the following parts: experimental samples, simulated heat source, drive device, measuring device and heat insulation device. The schematic diagram of the liquid cooling experiment is shown in Figure 17.

(1) The sample is made of aluminum. The model is divided into two symmetrical parts, which are then fastened together by screws, and a high-temperature epoxy resin-laminated sample is used to ensure tightness. The model is processed by professional workshops. In the process of machining, appropriate cutting speed, feed and cutting fluid were selected to ensure minimum roughness. Thermally conductive silica gel was used to fix the heat source on the surface of the sample (thermal conductivity of hot silicon is 2.1 W/m∙K). The experimental samples are shown in Figure 18.

(2) Thin-film resistors were taken as the analog heat source. This film resistor was MP9100 (Shenzhen Jinkena Electronics Co., Ltd., Shenzhen, China) with a section area of 11.5 × 14 mm^2^. The rated power of the film resistor is 100 W and the resistance was 20 Ω.

(3) The driving device is composed of a constant temperature water bath, a peristaltic pump, a water pipe, a multichannel direct current (DC) power supply and a measuring cylinder. The constant temperature water bath used in this experiment was a JULABO-VIVO RT2 (Qiwei Instrument Co., Ltd., Hangzhou, China), which kept the temperature at room temperature (20 °C). The peristaltic pump was a LHZW007 (United Zhongwei Technology Co., Ltd., Guangzhou, China). Transparent tubes with an inner diameter of 6 and an outer diameter of 8 mm were used to form the complete loop. The multichannel DC power supply (Tektronix/2230 G series) (Guce Power Supply, Guangzhou, China) was adopted in this work to supply the required input heat power.

(4) The measuring device was mainly composed of thermocouples, a temperature data acquisition instrument and a digital pressure meter. A temperature data acquisition instrument (Agilent 34970A) (Ruimanting Instrument Shop, Guangdong, China) was employed to record data. We used K-type thermocouples to measure the coolant temperatures at the inlet and outlet and the wall temperature. A digital pressure agent (Comark C9555) (Ximabaoxin Store, Shanghai, China) was used to measure the pressure drop between the inlet and outlet.

(5) Insulating tape was used to reduce heat loss to the surrounding environment of the radiator.

### 5.2. Experimental Methods and Procedures

(1) Twenty MP9100 film thermal resistors were evenly fixed on the surface of the heat sink substrate using thermal silica. We used a layer of thermal silica to eliminate the contact heat resistance between the film resistor and the heat sink. Figure 19 is the schematic diagram of the map of film thermal resistors.

(2) The heat exchanger is placed on the test bench.

(3) Four K-type thermocouples were inserted into four holes that were distributed sidewall of the heat exchanger to obtain the average wall temperature. At the same time, two K-type thermocouples were inserted into the inlet and outlet respectively to obtain the inlet and outlet liquid temperatures. The water pipes were, respectively, connected to the import and export of the minichannel cold plate, constant temperature water bath, peristaltic pump, and pressure gauge. Finally, the liquid cooling experiment platform was built. The experimental apparatus is shown in Figure 20.

Based on the method of Coleman et al. [34] and American Society of Mechanical Engineers (ASME) standard [35], we calculated the experiment uncertainty. The details of the uncertainty calculation is in our work [10]. The results are shown in Table 7.
(38)$$U_{R}={\left[\sum_{I=1}^{n}{\left(\frac{\partial R}{\partial V_{I}}U_{V_{I}}\right)}^{2}\right]}^{1/2}$$
where *U_Vi_* is the absolute error of the independent parameter and *n* is the number of variables. Therefore, we obtained the uncertainties listed in Table 7.

### 5.3. Experimental Data Analysis

When the entire experimental device was successfully connected, the flow was adjusted by the pump and the entire device ran. When all the data were stabilized, we recorded them.

Since the temperature of the channel wall could not be measured easily, we measured *T_b_* for convenience. The locations of the thermocouples are shown in Figure 19. The temperature of location A (*T_a_*) was derived by assuming one-dimensional heat conduction in the *y* direction [36]
(39)$$T_{a}=T_{b}-\frac{Qy}{L_{w}L_{w}k_{s}}$$
where *y* is 1.5 mm, and *T_b_* equals (*T**_a_*_1_ + *T**_a_*_2_ + *T**_a_*_3_ + *T**_a_*_4_)/4; and *T**_a_*_1_, *T**_a_*_2_, *T**_a_*_3_ and *T**_a_*_4_ are the temperature readings measured by the four thermocouples. Therefore, the wall temperature was obtained.
(40)$$T_{\mathrm{bar}}=\frac{\sum_{i=1}^{4}T_{ai}}{4}$$

To ensure the stability of the experimental data, the fluctuation of the measured data had to be less than 0.1% (e.g., the temperature reading range was less than 0.1 K). The measured data records are shown in Table 8 as well as a comparison of the experimental data and simulation data.

From Table 8, we found that both $T_{bar}$ and $T_{g}$ decreased when the flow rate increased. The change trend of the measured temperature was similar to that of the simulated temperature. When the flow rate was 59.4 mL/s, the error between the measured the $T_{bar}$ and the simulated $T_{bar}$ was 2.6 K, and the error between the measured $T_{g}$ and the calculated $T_{g}$ was 1.4 K.

On the basis of the measured results, we further analyzed the experimental and simulated Nusselt numbers for different Reynolds numbers. Figure 21 shows that both Nusselt numbers increased with the increase in the Reynolds number, and the maximum error between them did not exceed 5%. This indicated that the experimental results were in good agreement with the numerical results.

Figure 22 shows that the pressure drop gradually increased as the Reynolds number increased. The variation of the measured pressure drop was similar to that of the numerical pressure drop. The experimental pressure drop was always larger than that of the simulated pressure drop. The maximum error did not exceed 0.2%, which verified the correctness of the numerical results.

## 6. Conclusions

To improve the heat transfer rate and the uniformity of the substrate bottom temperature, we proposed a new type of heat exchanger design with three inlets and one outlet by using topology optimization. The thermal performance of heat sinks designed by topology optimization were studied numerically and experimentally. The main conclusions are as follows:Compared with conventional straight channel radiators, the two heat sinks had better heat dissipation performance due to the secondary channels because of interruptions achieved by topology optimization.The heat sink designs obtained by topology optimization had lower thermal resistance compared to conventional straight channel radiators.Model-2 was the best design for flow and heat transfer performance.Channel depth had a significant effect on the heat sink designed by topology optimization.A critical channel depth exists for obtaining the best flow and thermal performance.The temperature difference of Model-1 was reduced by 37.5% and that of Model-2 by 62.5% compared to Model-3.Compared with Model-3, the thermal resistance of Model-1 was reduced by 21.86%, while that of Model-2 was reduced by 47.99%.The error between the simulated results and experimental results was no more than 10%, which verified the accuracy of numerical results.

## Figures and Tables

**Figure 1 micromachines-12-00594-f001:**
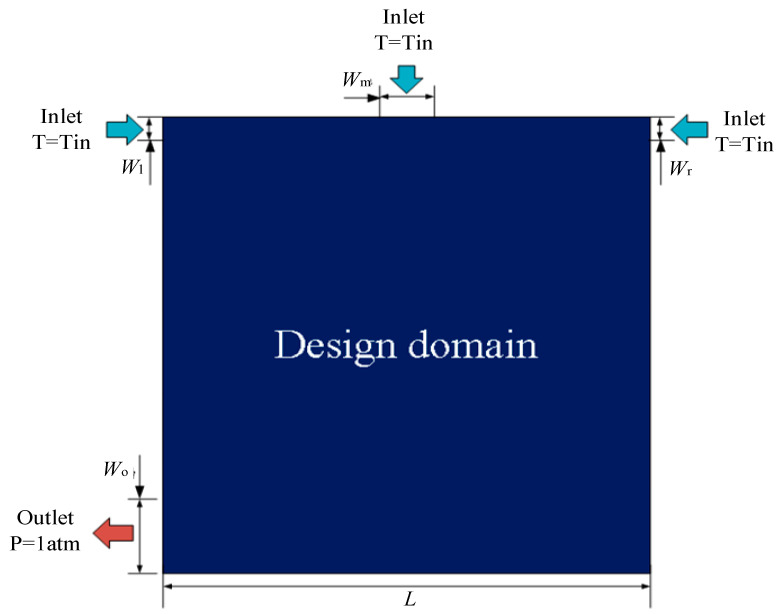
Topology optimization design model.

**Figure 2 micromachines-12-00594-f002:**
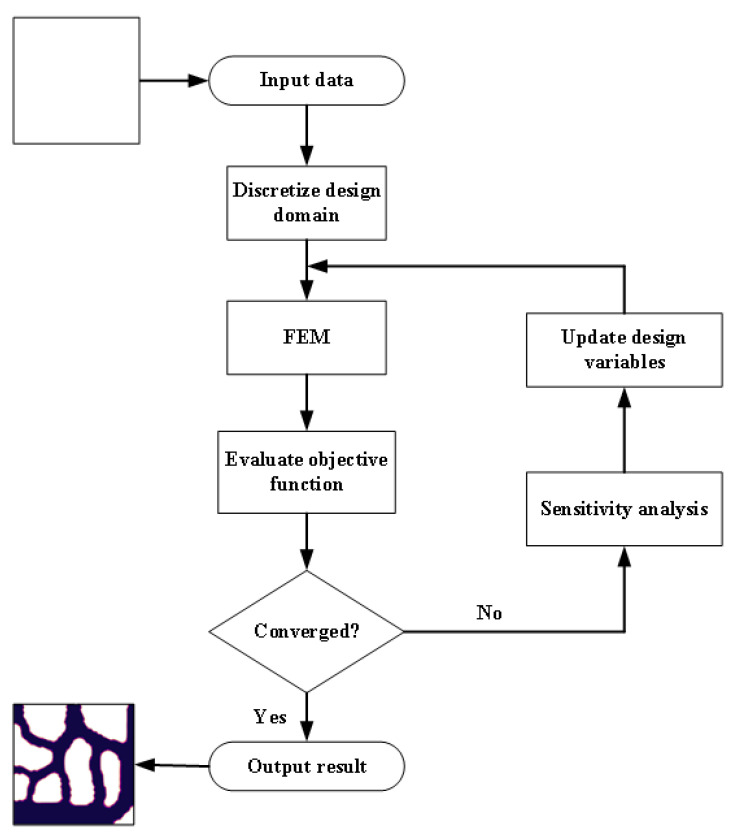
Topology optimization flowchart.

**Figure 3 micromachines-12-00594-f003:**
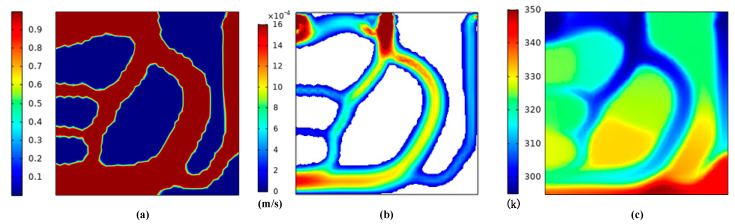
Topology optimization contours for the lowest average temperature as the goal (**a**) the average material volume factor (**b**) the velocity field (**c**) the temperature field.

**Figure 4 micromachines-12-00594-f004:**
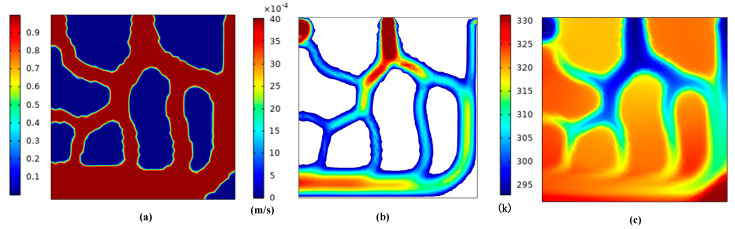
Topology optimization contours for the minimum temperature difference as the goal (**a**) the average material volume factor (**b**) the velocity field (**c**) the temperature field.

**Figure 5 micromachines-12-00594-f005:**
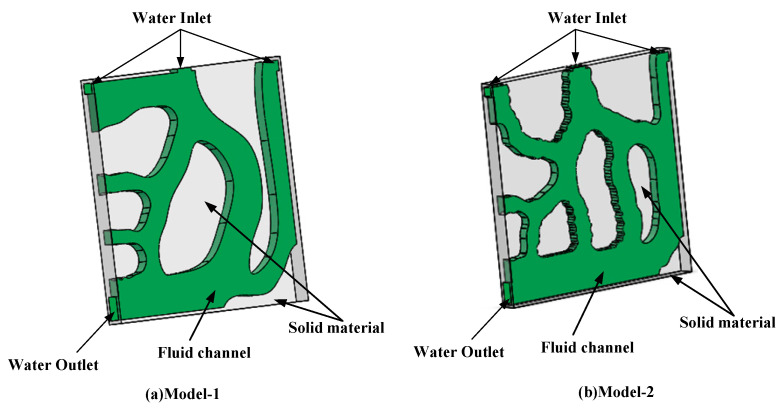
3D model diagram of (**a**) Model-1 and (**b**) Model-2.

**Figure 6 micromachines-12-00594-f006:**
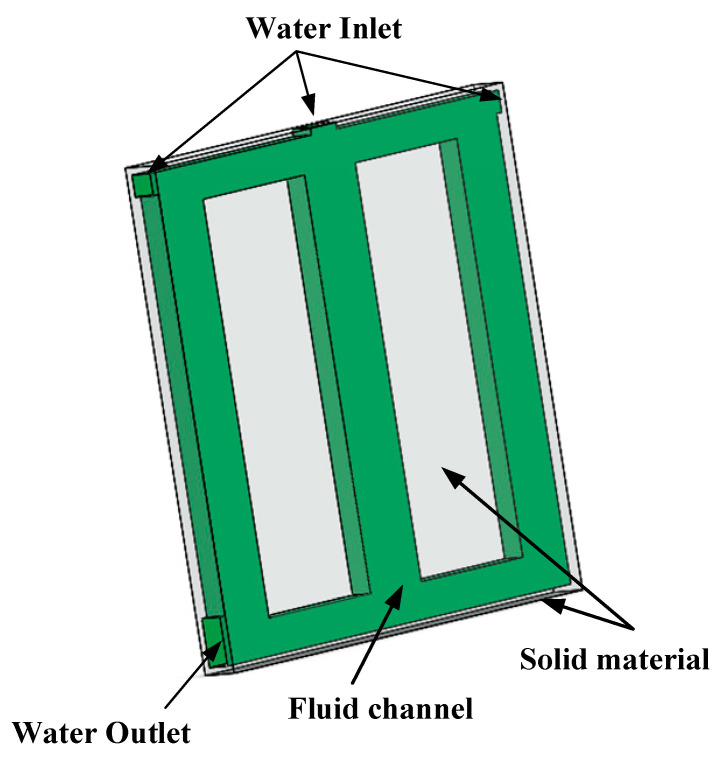
3D model diagram of Model-3.

**Figure 7 micromachines-12-00594-f007:**
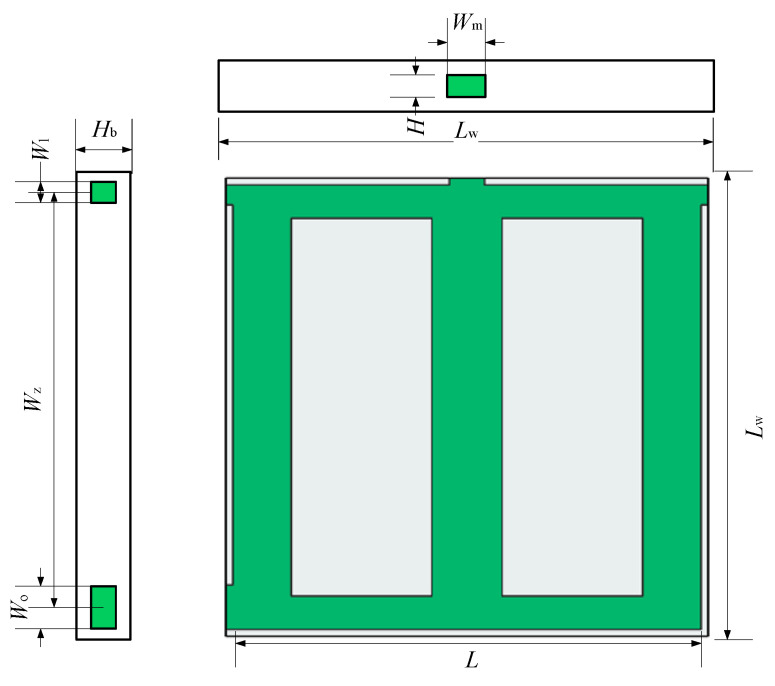
Three views of Model-3.

**Figure 8 micromachines-12-00594-f008:**
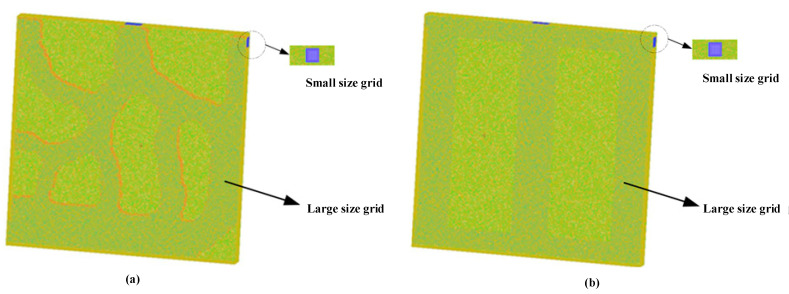
Grid layout of (**a**) Model-2; (**b**) Model-3.

**Figure 9 micromachines-12-00594-f009:**
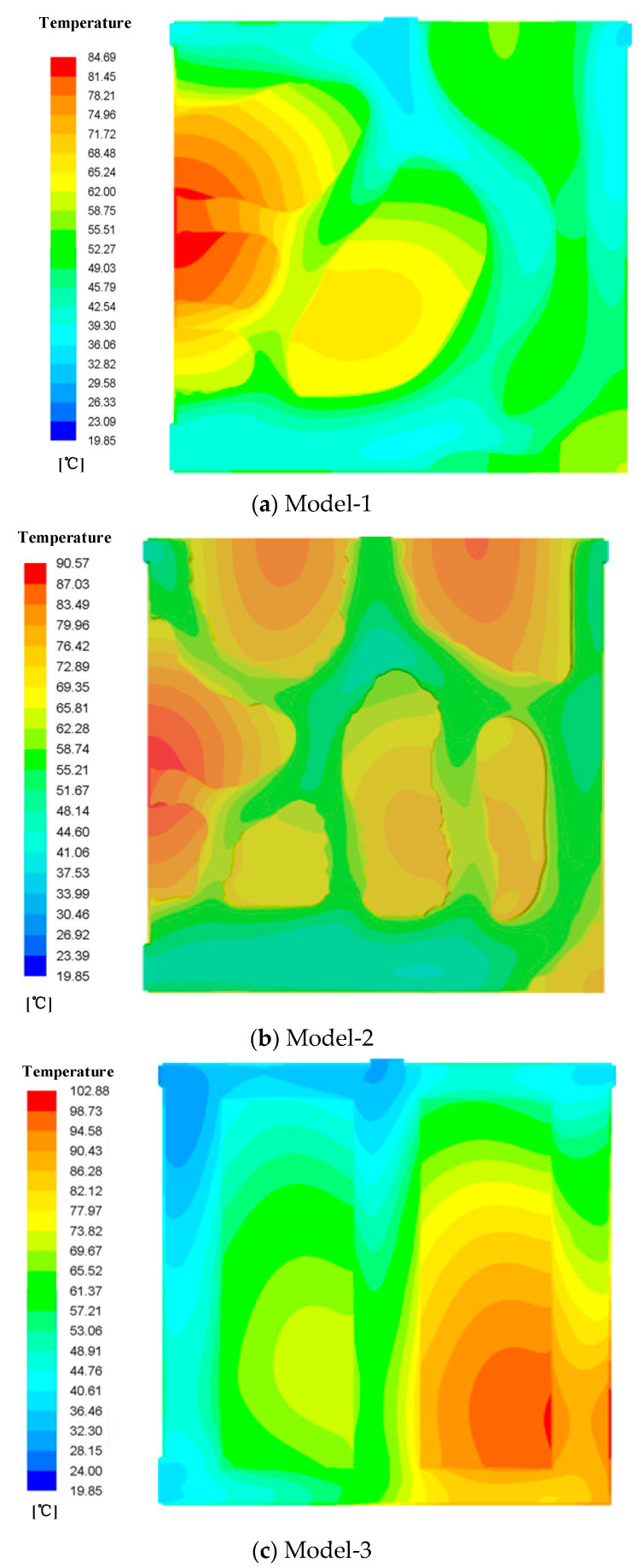
The substrate temperature contours for different Models. (**a**) Model-1; (**b**) Model-2; (**c**) Model-3.

**Figure 10 micromachines-12-00594-f010:**
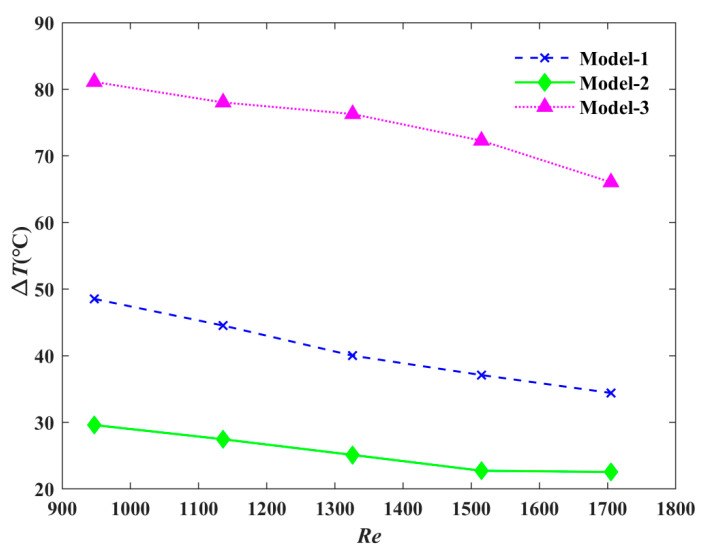
The influence of Reynolds number on temperature difference.

**Figure 11 micromachines-12-00594-f011:**
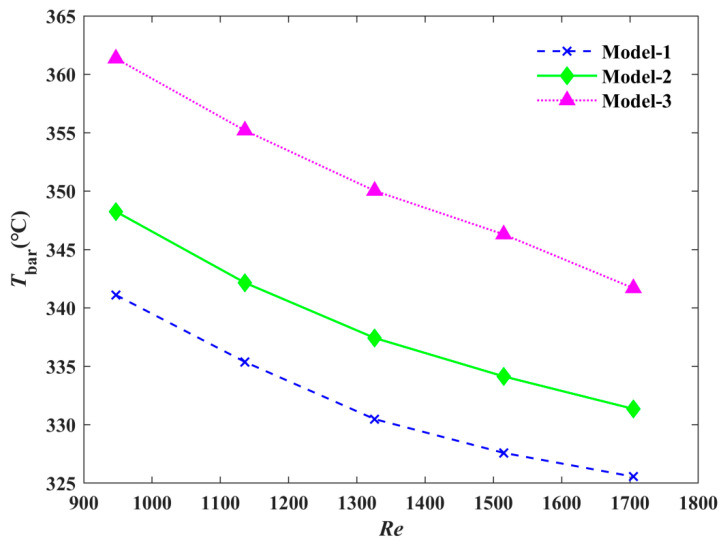
The effect of the Reynolds number on average temperature.

**Figure 12 micromachines-12-00594-f012:**
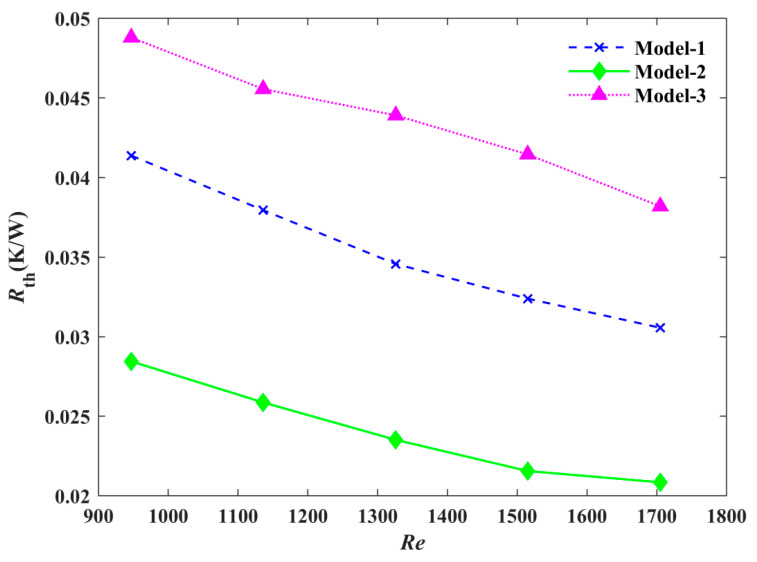
Comparison of the thermal resistance of the three models under different Reynolds numbers.

**Figure 13 micromachines-12-00594-f013:**
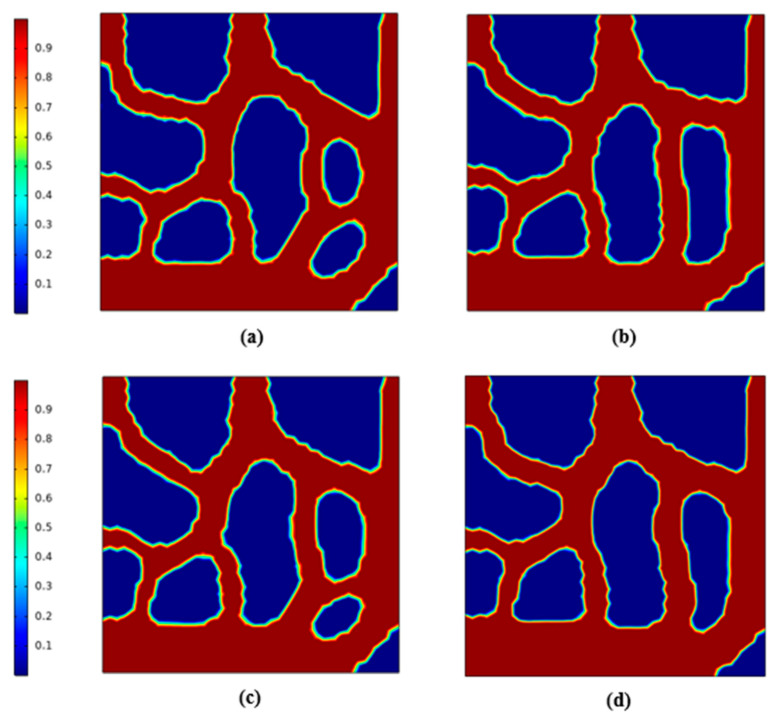
Topology optimization contours for the minimum temperature difference as the goal, (**a**) depth = 4 mm; (**b**) depth = 6 mm; (**c**) depth = 8 mm; (**d**) depth = 12 mm.

**Figure 14 micromachines-12-00594-f014:**
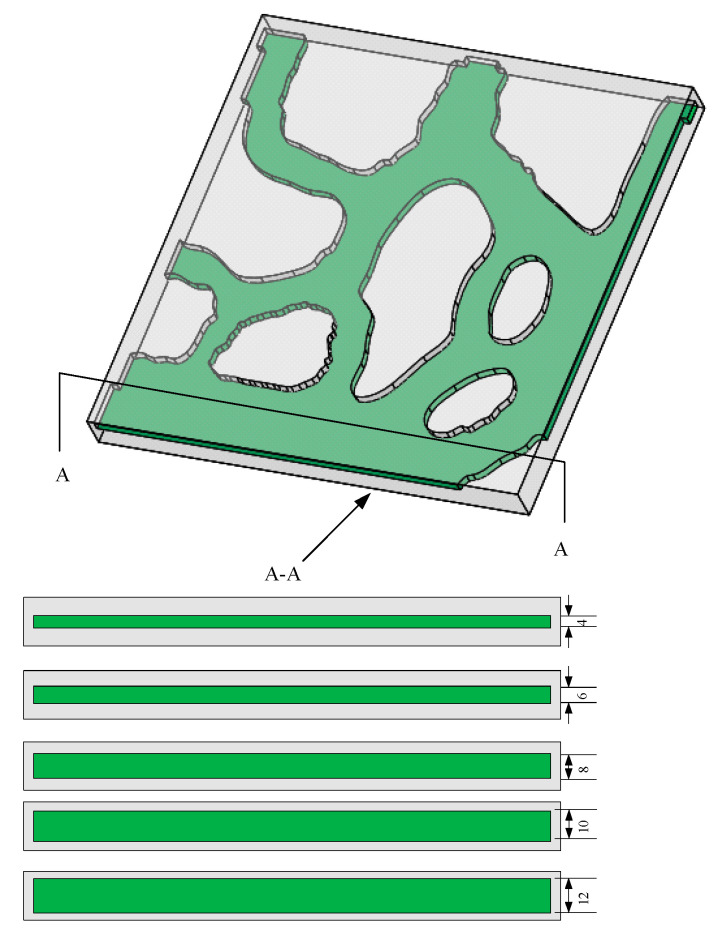
3D model.

**Figure 15 micromachines-12-00594-f015:**
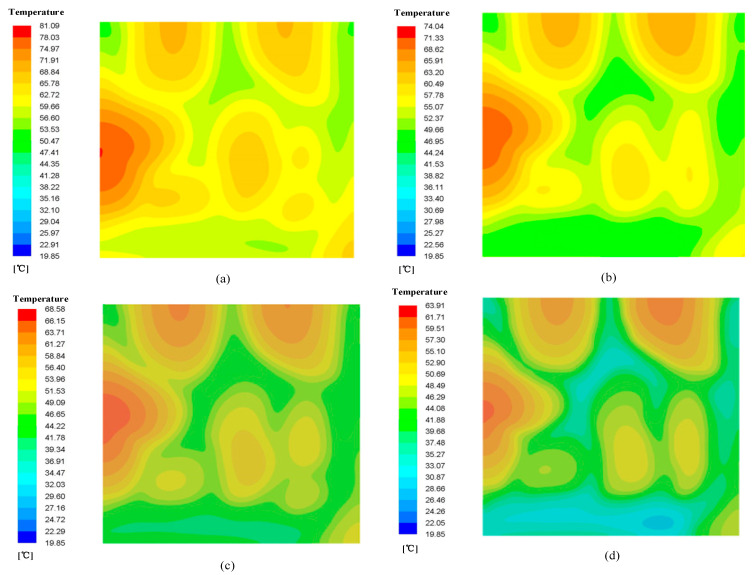
Temperature contours of heat exchangers: (**a**) depth = 4 mm; (**b**) depth = 6 mm;(**c**) depth = 8 mm;(**d**) depth = 12 mm.

**Figure 16 micromachines-12-00594-f016:**
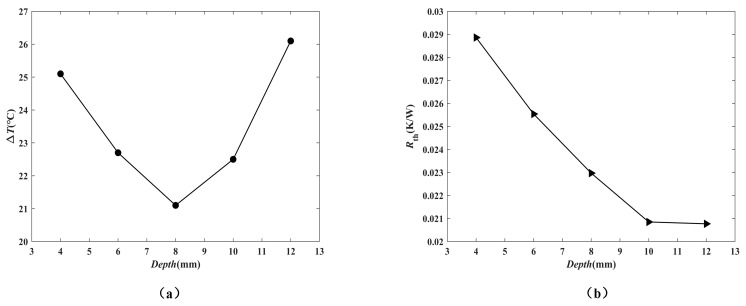
The effect of channel depth on (**a**) the temperature difference of the substrate surface of the exchanger; (**b**) the thermal resistance; (**c**) Nusselt number; (**d**) the pressure drop.

**Figure 17 micromachines-12-00594-f017:**
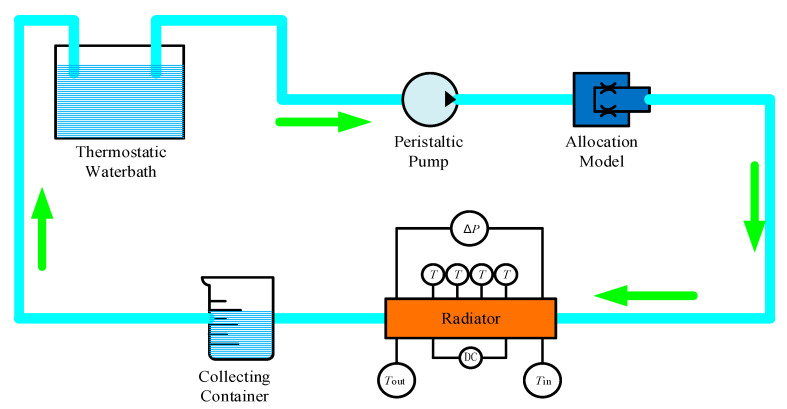
The schematic of the experiment.

**Figure 18 micromachines-12-00594-f018:**
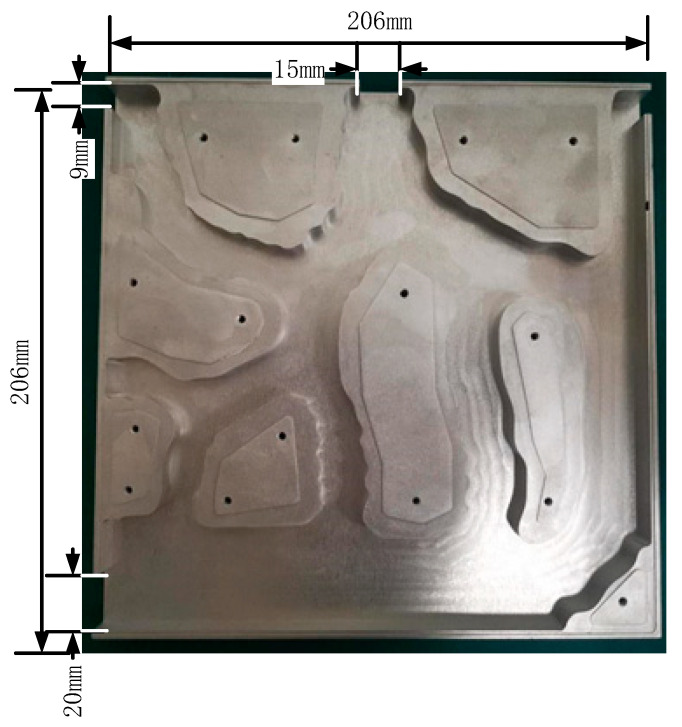
The test simple of Model-2 model.

**Figure 19 micromachines-12-00594-f019:**
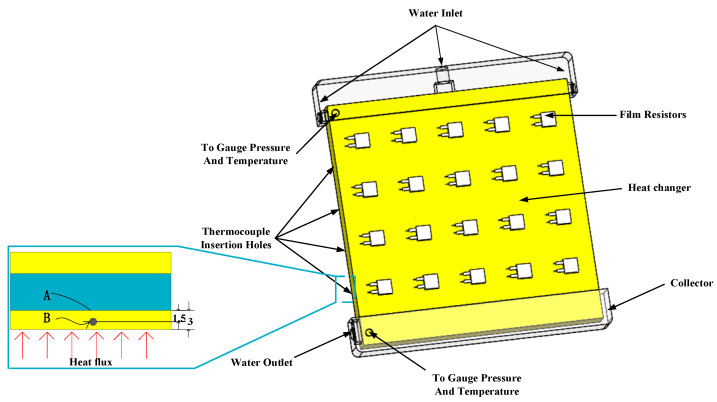
Assembly drawing of heat exchanger and film thermal resistance.

**Figure 20 micromachines-12-00594-f020:**
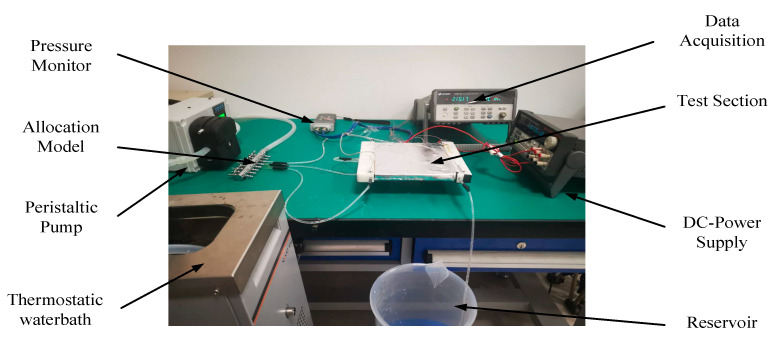
Testing apparatus for the performance evaluation.

**Figure 21 micromachines-12-00594-f021:**
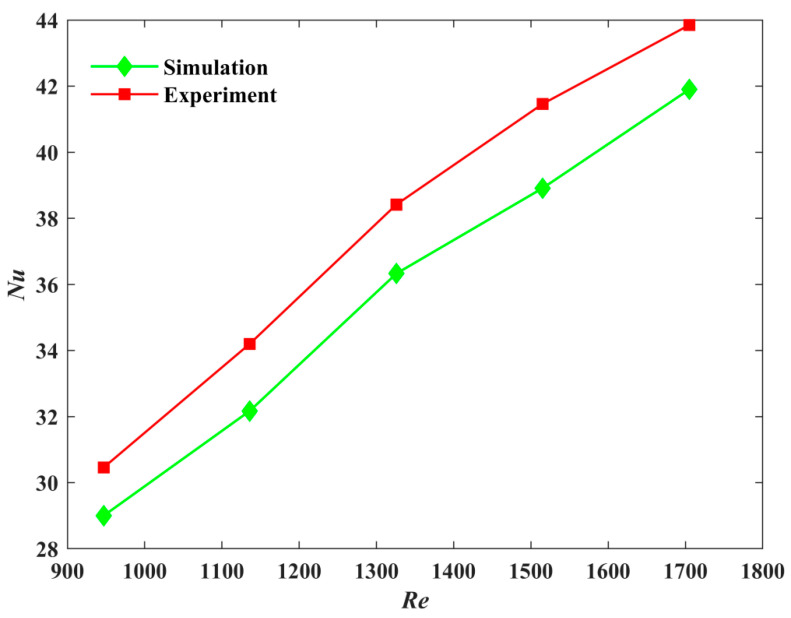
Comparison of Nusselt numbers between Model-2 Experiment and Simulation.

**Figure 22 micromachines-12-00594-f022:**
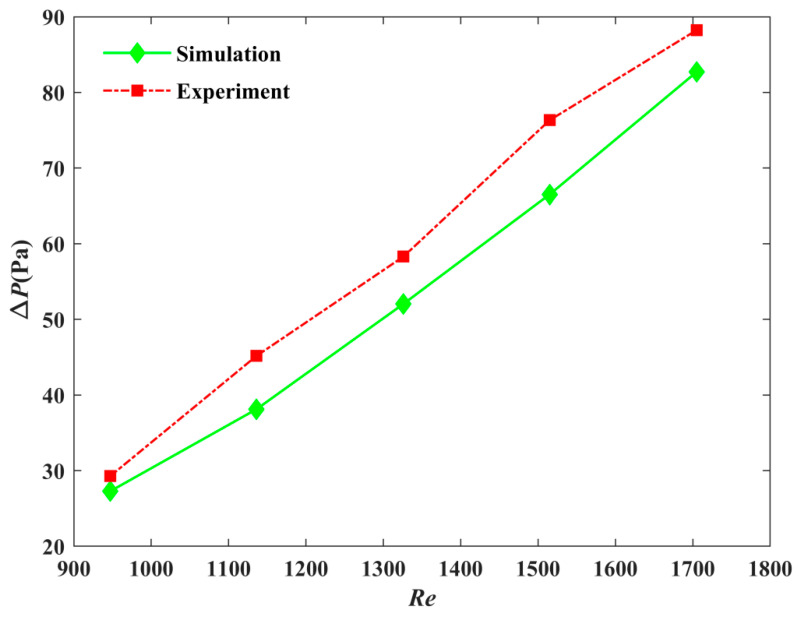
Comparison of the pressure drop between experiment and simulation for Model-2.

**Table 1 micromachines-12-00594-t001:** Parameters of the design domain.

Dimension	Size (mm)
*L*	200
*W*_l_, *W*_m_, *W*_r_, *W*_o_	9, 15, 9, 20

**Table 2 micromachines-12-00594-t002:** The boundary and parameters of topology optimization.

Variable	Value	Variable	Value
*Q* (W)	60	*p_α_*	10^−2^
*u* (m/s)	0.002	*p_k_*	10^−2^
*α_s_* (Pa·s/m^2^)	10^6^	*p_ρ_*	10^−2^
*α_f_* (Pa·s/m^2^)	0	*p_C_*	10^−2^
*μ* (Pa·s)	10^−3^	*q*	10^−3^
*h_t_*	1	*w*	0.5
*f_v_*	50%	*γ*	0.5

**Table 3 micromachines-12-00594-t003:** Material properties of the topology optimization.

Material	*k* (W/(m·K))	*C_p_* (J/kg·K)	*ρ* (kg/m^3^)	*μ* (Pa·s)
Water	0.61	4180	1000	0.001
Aluminum	237	900	2700	

**Table 4 micromachines-12-00594-t004:** Three-dimensional model structure parameters.

Dimension	Size (mm)
*L*_w_ × *L*_w_ × *H*_b_	206 × 206 × 16
*W*_m_ × *H*	15 × 10
*W*_l_ × *H*	9 × 10
*W*_o_ × *H*	20 × 10
*W*_z_, *L*	171,200

**Table 5 micromachines-12-00594-t005:** The Reynolds number and volume flow rate corresponding to the speed.

*Q_v_* (mL/s)	*u* (m/s)	*Re*
33	0.1	947
39.6	0.12	1136
46.2	0.14	1326
52.8	0.16	1515
59.4	0.18	1705

**Table 6 micromachines-12-00594-t006:** Grid independence verification.

	Model-2			Model-3		
*Grids*	5,536,079	9,561,361	18,450,641	5,541,460	9,569,202	13,299,555
*T_max_* (K)	337.5	337.2	335.4	371.1	374.0	374.1
*Error* (%)	0.085	-	0.556	0.785	-	0.021
Δ*P* (Pa)	88.9	91.3	92.8	107.1	103.5	107.0
*Error* (%)	2.583	-	1.640	3.417	-	3.298

**Table 7 micromachines-12-00594-t007:** Uncertainties of the experimental parameters.

Parameter	Absolute Uncertaintie	Reletive Uncertainitie
*Channel height (H)*	±0.05 mm	-
*Channel width (W)*	±0.05 mm	-
*Temperature (T)*	±0.1 K	-
*Volumetric flow rate (Q_v_)*	-	±10%
*Pressure drop* (Δ*P*)	-	±0.34%
*Nusselt number (Nu)*	-	±9.09%

**Table 8 micromachines-12-00594-t008:** Comparison of the measured and simulated temperature.

Q*_v_* (mL/s)	Ex-*T_bar_* (K)	Si-*T_bar_* (K)	Ex-*T_g_* (K)	Si-*T_g_* (K)
33	352.3	341.1	310.5	300.3
39.6	345.3	335.4	306.7	299.1
46.2	336.5	330.5	302.6	298.2
52.8	331.8	327.6	299.8	297.6
59.4	328.2	325.6	298.6	297.2

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
