# Peer review of "Study of the Performance of a Novel Radiator with Three Inlets and One Outlet Based on Topology Optimization"

_micromachines, 2021, doi:10.3390/mi12060594_

Round 1

Reviewer 1 Report

Dear authors,

Please update the manuscript with the comments suggested below to improve the quality of the manuscript.

  1. The abstract also needs to be incorporated with the gist of the complete work in the manuscript. In the current version of the abstract, only the summary of the work is mentioned in a very broad view. Instead please include specific details of the research study presented in the manuscript.
  2. In the abstract, please include the details of the limitations of the current/previous technologies and specify the scientific advances made in the proposed novel radiator with three inlets and one outlet using topology optimization theory, to overcome those limitations.
  3. Please update the abstract with the experimental conditions such as the temperatures of the measured quantities and corresponding temperature differences and heat transfer rates that need to be incorporated.
  4. In the abstract, In lines 19-20, kindly incorporate the quantification of results to have the reader understand the magnitude of the results. For example: in lines 19-20, ..it was mentioned that, the numerical results agree well with the experimental results. Please include the details of the numerical results and experimental results for comparison.
  5. The abstract of a scientific research paper should be precisely mentioning the specific research question that is answered, experimental conditions, operational parameters, results, and conclusions. Please explain and provide the specifications of the scientific questions answered in the proposed study.
  6. In the introduction, please include more details of the peer studies performed on the ‘radiators with multiple inlets and outlet heat exchangers using topology optimization’, to guide the reader to understand the importance of the research study performed. Also, include corresponding references in the text when mentioning the details.
  7. In the introduction, please include the knowledge gaps existing in the current research work and prior studies performed in the field. Very importantly, please specify the need for the current work presented in the manuscript.
  8. In the last paragraph of the introduction, kindly include the details of the broader impacts on the study made and the results achieved. It is very important to provide the future scope of the research performed to make a strong impact on the readers on the research performed/Study proposed.
  9. Please update the figures (figure -5,6,7,8) with high-quality images.
  10. In the section-5 section, kindly incorporate the logical reasoning and scientific conclusions made from the plots in the figures, and, also please use the ongoing research results of peers with appropriate references to support your arguments and statements.
  11. In the figure-22, please include the scale to the image for the reviewer to understand the size of the model-2.
  12. Please revise the manuscript with English grammar. There are many places that the manuscript needs to be improved with respect to English writing.

Reviewer 2 Report

  1. Please remove any first plural form and substitute with either impersonal or passive ones
  2.  In the abstract Model-1, -2, -3 are not referred to. The reader cannot understand (even with only a word) which model authors are speaking of. If you don't succeed in indicating the model, you should speak only of three different models.
  3. The bibliography could be enriched with reference to new Heat Exchangers models coming out from the scientific environment. The following publications could conveniently find a place among your references:
    1. A hybrid method for the cross flow compact heat exchangers design, Applied Thermal Engineering, 2017, 111, pp. 1129–1142

    2. The design of countercurrent evaporative condensers with the hybrid method, Applied Thermal Engineering, 2018, 130, pp. 889–898

    3. The hybrid method applied to the plate-finned tube evaporator geometry, International Journal of Refrigeration, 2018, 88, pp. 67–77

  4. For the Delta p trend description, authors should try to use an adimensional parameter.
  5. Figures from 11 to 13 should higher their quality and use the same line style (please, use a higher differentiation among lines) for the same model. 
  6. Figures from 17 to 20 could be reduced at two with double vertical scales and a better choice of the vertical axis limit should be chosen to highlight differences. Other more compact graph configurations could also be used. 
  7. Figure 21 should be of higher quality.
  8. Conclusions should underline some short issues about originality and novelty of the paper.
  9. English could be better cared for. Ask collaboration to an English  mothertongue speaker  
  10. Considering the interest in a practical application, author should evaluate if the better unit to show results of temperature onare  graphs celsius, in place of kelvins.
  11. Page 26, row 14 - It seems that a part of the paper was canceled  and something is still there.

Sincerely

the Reviewer

Reviewer 3 Report

The manuscript investigates the thermal performance of a proposed radiator with three inlets and an outlet designed by topology optimization. Authors compared the thermal and flow performance of the three different models by means of  FEM numerical method via COMSOL Multiphysics and finally they conducted an experiment for a model with best temperature uniformity.

The topic is interesting and fits well into the journal's scope.The manuscript is well-written and organized and delivers what claims in abstract. The Introduction communicates state-of-the-art effectively and numerical model has been elaborated and validated comprehensively. Moreover, the experimental procedures has been clearly discussed and obtained results agree with numerical ones. The presented  remarks in conclusions are also supported by obtained results.

According to these remarks, I believe the manuscript is eligible to be endorsed for publication.

Round 2

Reviewer 1 Report

Dear authors,
Thank you for updating the manuscript with recommended changes.

Reviewer 2 Report

Authors made what they were requested.

The paper can be now published after some minor changes.

Nusselt and Reynolds (adimensional) numbers abbreviations (in formulas and in the nomenclature) should not be written with the second letter as a subscript. Nu and Re (the right form is in the figures of your paper) are the first two letters of the surnames of excellent historical persons studying thermodynamics ... and that's it.